# A Wound Environment Control System to Avoid Major Amputation in Diabetic Foot Ulcers

**DOI:** 10.3390/medicina59030430

**Published:** 2023-02-22

**Authors:** Julien Vouillarmet, Gisele Fribourg, Lauriane Labaisse, Nellie Della-schiava

**Affiliations:** 1Hospices Civils de Lyon, Service d’Endocrinologie, Diabète et Nutrition, Centre Hospitalier Lyon-Sud, 69495 Pierre Bénite, France; 2Hospices Civils de Lyon, Service de Chirurgie Vasculaire, Hôpital Edouard Herriot, 69003 Lyon, France

**Keywords:** medical device, diabetic foot ulcer, amputation, wound environment

## Abstract

We report the case of a 58-year-old patient with a diabetic foot lesion at high risk of major amputation successfully treated by a new innovative wound environment control system.

## 1. Introduction

Despite improvements in its prevention and treatment, diabetic foot syndrome remains a major cause of non-traumatic amputation worldwide [1]. Factors related to the wound environment, such as moisture, temperature, and oxygen, are important in accelerating wound closure and decreasing the risk of amputation. In 1962, Winter developed the concept of “Moist Wound Healing”, demonstrating that moisture plays a central role in wound management [2]. High moisture improves oxygen transfer [3] and epithelialization, and hence, wound healing. Concerning temperature, while values lower than 32–33 °C induce delayed migration of fibroblasts and epithelial cells, higher temperatures of around 34–36 °C stimulate fibroblasts’ growth and oxygen pressure due to vasodilation [4]. Concerning oxygen delivery, an increase in transcutaneous oxygen levels, as well as alternating controlled phases of hypoxia and hyperoxia, can improve fibroblast proliferation and angiogenesis. A recent study reported that daily Cyclical Topical Wound Oxygen Therapy (TWO2^®^) significantly improves the rate of healing in hard-to-heel superficial diabetic foot ulcers without infection or arteriopathy [5].

Vistacare^®^ (DTAMedical SAS, Loulle, France) is a device developed to control the wound environment in each phase of the healing process (Figure 1). It is divided in two parts: a generator and a closed chamber, where the leg presenting with the diabetic foot ulcer is placed. Moisture (from 40 to 85%), temperature (from 32 to 34 °C), and oxygen levels (from 21 to 50%) are controlled and modulated in three predetermined sets classified as A (for debridement), B (for granulation), and C (for epithelialization). Temperature is controlled between 32 and 34 °C for all sets. For set A, the average moisture is 80 ± 10% with variations between the two extremes, and the average oxygen level is 40 ± 10% with progressive variation. For set B, the average moisture is 60 ± 20% with variations between the two extremes, and the oxygen level varies between 20% and 50%, with brutal and cyclic variations in stress cells. For set C, the average moisture is 50 ± 10% with variations between the two extremes, and the average oxygen level is 30 ± 10% with progressive variation. No dressing is needed during the entire treatment, with the wound being in a controlled environment. A window at the top of the chamber allows for direct visual access to the wound. Depending on the evolution of the wound, it is possible to change from one predetermined set to another at any time during the treatment. Each (+) sign displayed on the screen within the set pictogram corresponds to 3 h of treatment within the 12 h cycle, which is repeated until human intervention occurs (Figure 1B). Mechanical debridement can also be performed when needed, and offloading of the wound is maintained during the entire treatment. When using Vistacare^®^, patients should be hospitalized for 10 to 14 days, and the wound environment control treatment should be applied for at least 16 h per day.

## 2. Case Presentation

We report herein the case of a 58-year-old woman admitted to our unit for a post-amputation diabetic foot ulcer. She was diagnosed with diabetes in 2014. There was no argument for type 2 diabetes (no familial history of diabetes, no history of overweight, no metabolic syndrome). Her BMI at diagnosis was 22 kg·m^−2^. The antibodies for type 1 diabetes were absent. The pancreas was atrophic on a CT scan, without history of pancreatitis crisis, alcohol abuse, or lithiasis. In this context, and given the high blood glucose level with ketosis, insulin treatment was proposed. Due to the atrophic aspect of the patient’s pancreas and her normal weight, GLP-1 agonists were not introduced. The diabetes was complicated by distal polyneuropathy, lower limb arteriopathy, and coronary artery disease. The patient never smoked.

In July 2020, the patient developed a new wound on her right first toe related to the rubbing of her shoes, which was secondarily infected. Due to osteomyelitis and necrosis, a first ray amputation associated with endoluminal lower limb revascularization was performed in August. Necrosis and severe arterial stenosis were found on angiography (Figure 2A). This unfavorable outcome led to a transmetatarsal amputation on 14 January 2021 (Figure 2B) followed by a Lisfranc amputation on 19 January 2021, associated with prolonged angioplasty of each leg artery with good angiographic control.

Bone samples on healthy bone margins were collected during surgery for bacterial and histological analyses. Staphylococcus meti-S, Enterococcus faecalis, and Corynebacterium striatum were identified. The histological analysis confirmed acute osteomyelitis. A combination of cefoxitin and vancomycin was started, and then, adapted with rifampicin and vancomycin for a 6-week course.

The patient was admitted to our diabetic foot unit on 26 January 2021 due to an unfavorable post-surgery clinical outcome with extended necrosis despite negative wound pressure therapy, offloading, and adapted antibiotic treatment to avoid transtibial amputation.

Upon admission, renal function was normal. Metabolic control was perfect with HbA1c at 5.9%, C-LDL was at 1.1 nmol/L under statin treatment, and blood pressure was controlled. PCR was stable at 21 mg/L with a normal neutrophil count.

Wound measurement was 63 × 42 × 36 mm (length × width × depth; Figure 2C). The wound was covered by fibrin and reported to be painful during debridement. The wound was classified as SINBAD 4 and IDSA 3o for infection, according to the IWGDF guidelines [6].

Wound environment control treatment using Vistacare^®^ was started. The initial program applied aimed to enhance debridement by exposing the wound to high levels of moisture. The predetermined set A was thus mainly used for the first week of treatment. On Day-3, the slough was removed after light mechanical debridement without inducing any pain (Figure 2D). During the second week of treatment, both the A and B sets were applied equally to further induce debridement and stimulate granulation. Offloading was maintained during the entire treatment.

After 2 weeks of treatment, the wound showed healthy granulation tissue (Figure 2E). A wound dressing was applied until a skin graft was performed on 10th March (Figure 2F). Complete epithelialization of the wound was obtained on 15th June (Figure 2G,H). In August, the patient could walk normally using therapeutic shoes.

## 3. Conclusions

Vistacare^®^ is a new innovative device for wound management which is currently being assessed in patients with acute and chronic leg wounds in an open-label, prospective, multi-center study (NCT03790202). However, this is the first report concerning the use of this specific wound environment control system in the management of a patient with a severe diabetic foot ulcer, which allowed us to avoid major amputation. This device should be considered as complementary to usual treatment, notably for ischemic wounds, when standards of care are insufficient and when the use of a topical wound negative pressure device is painful and/or ineffective. Additional reports are needed to better define its use in diabetic foot management.

## Figures and Tables

**Figure 1 medicina-59-00430-f001:**
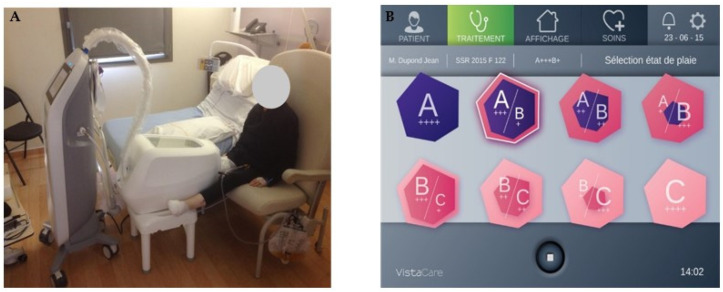
Vistacare^®^ device. (**A**) The generator and the chamber where the leg with the wound is placed; (**B**) control screen of the Vistacare^®^ device displaying the different pictograms for each set (a (+) sign corresponds to 3 h of treatment within a 12-h cycle).

**Figure 2 medicina-59-00430-f002:**
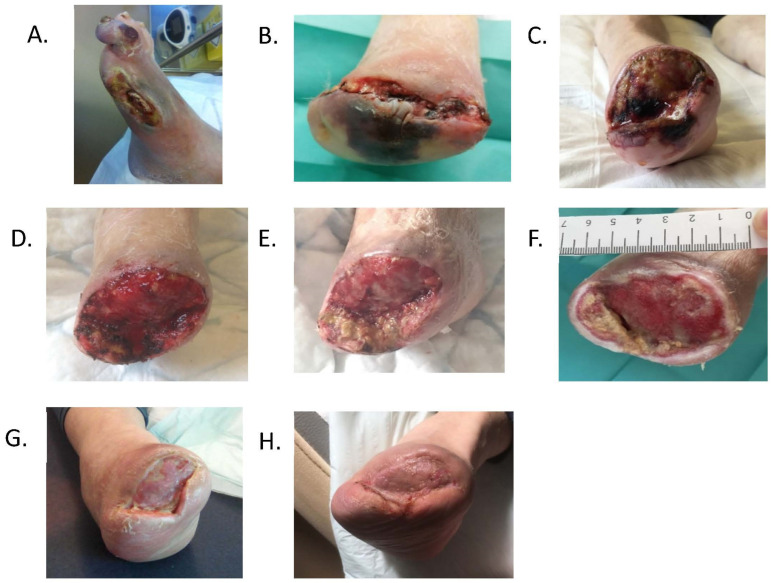
Wound management following diabetic foot ulcer-related amputation. (**A**) Four months after first ray amputation; (**B**) one day after transmetatarsal amputation; (**C**) upon admission to the diabetic foot unit one week after Lisfranc amputation (26 January 2021); (**D**) three days after applying the wound environment control system Vistacare^®^; (**E**) two weeks of treatment using Vistacare^®^; (**F**) before skin graft; (**G**) six weeks after skin graft; (**H**) three months after skin graft.

## Data Availability

The data that support the findings of this study are available from the corresponding author.

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
