# Peer review of "A Wound Environment Control System to Avoid Major Amputation in Diabetic Foot Ulcers"

_medicina, 2023, doi:10.3390/medicina59030430_

Round 1

Reviewer 1 Report

The Authors report the case of a patient with an ischemic/infected post amputation diabetic foot ulcer treated by Vistacare device.

The manuscript need major revision. 

1) It is not clear how the device works. Authors should provide a more extensive description of Vistacare mechanism of action. Provide a detailed description of both A and B sets; how granulation is stimulated?

2) In the case description authors should specify the sex, duration of diabetes, ulcer duration.

3) Why a young women (58 years) with HbA1c 5.9%, affected by vascular disease (coronary artery disease, peripheral arterial disease) was treated by insulin? According to International guidelines, she may benefit of cardiovascular protective of non insulin drugs.

4) What do Authors mean by "unfavorable outcomes"? Why lower limb revascularization was unfavorable and how you define that? Why patient did not undergo to a new revascularization? 

5) Ulcer grading according to IWGDF Guidelines need to be defined.

6) How author define infection? How a specimen for culture has been collected? What is/are the pathogens isolated and which antibiotic therapy has been administered.  

7) Did you use a pain scales?

8) The authors define the device as a possibile treatment for ischemic wound. How you define ischemia? Please specify that it should be useful after the standard of care application.  

Reviewer 2 Report

Congratulations for the authors, The manuscript "A wound environment control system to avoid major amputation in diabetic foot ulcer"  It is very interesting. I would just like to clarify some aspects that I have not clear.

Was a microbiological cultive and was the patient treated with antibiotics therapy?

Is it in the diabetic foot care a podiatry ?  who perform the cures and discharges of the foot, the podiatry?

Does the vistacare medical device have a maximum time to use? In which situtation  the treatment is suspended? under which criteria?

Thanks again, I was pleasantly surprised by this work.

Round 2

Reviewer 1 Report

Thank you for your revisions